# THE VARIATIONAL DEFICIENCY BOTTLENECK

## ABSTRACT

We introduce a bottleneck method for learning data representations based on channel deficiency, rather than the more traditional information sufficiency. A variational upper bound allows us to implement this method efficiently. The bound itself is bounded above by the variational information bottleneck objective, and the two methods coincide in the regime of single-shot Monte Carlo approximations. The notion of deficiency provides a principled way of approximating complicated channels by relatively simpler ones. The deficiency of one channel w.r.t. another has an operational interpretation in terms of the optimal risk gap of decision problems, capturing classification as a special case. Unsupervised generalizations are possible, such as the deficiency autoencoder, which can also be formulated in a variational form. Experiments demonstrate that the deficiency bottleneck can provide advantages in terms of minimal sufficiency as measured by information bottleneck curves, while retaining a good test performance in classification and reconstruction tasks.

*Keywords:* Variational Information Bottleneck, Blackwell Sufficiency, Le Cam Deficiency, Information Channel

## 1 INTRODUCTION

The information bottleneck (IB) is an approach to learning data representations based on a notion of minimal sufficiency. The general idea is to map an input source into a representation that retains as little information as possible about the input (*minimality*), but retains as much information as possible in relation to a target variable of interest (*sufficiency*). See Figure 1. For example, in a classification problem, the target variable could be the class label of the input data. In a reconstruction problem, the target variable could be a denoised reconstruction of the input. Intuitively, a representation which is minimal in relation to a given task, will discard nuisances in the inputs that are irrelevant to the task, and hence distill more meaningful information and allow for a better generalization.

In a typical bottleneck paradigm, an input variable $X$ is first mapped to an intermediate representation variable $Z$, and then $Z$ is mapped to an output variable of interest $Y$. We call the mappings, resp., a representation model (*encoder*) and an inference model (*decoder*). The channel $\kappa$ models the true relation between the input $X$ and the output $Y$. In general, the channel $\kappa$ is unknown, and only accessible through a set of examples $(x^{(i)}, y^{(i)})_{i=1}^{N}$. We would like to obtain an approximation of $\kappa$ using a probabilistic *model* that comprises of the *encoder-decoder pair*.

The IB methods (Witsenhausen & Wyner, 1975; Tishby et al., 1999; Harremoës & Tishby, 2007; Hsu et al., 2018) have found numerous applications, e.g., in representation learning, clustering, classification, generative modeling, model selection and analysis in deep neural networks, among others (see, e.g., Shamir et al., 2008; Gondek & Hofmann, 2003; Higgins et al., 2017; Alemi et al., 2018; Tishby & Zaslavsky, 2015; Shwartz-Ziv & Tishby, 2017).

In the traditional IB, minimality and sufficiency are measured in terms of the mutual information. Computing the mutual information can be challenging in practice. Various recent works have formulated more tractable functions by way of variational bounds on the mutual information (Chalk et al., 2016; Alemi et al., 2016; Kolchinsky et al., 2017), sandwiching the objective function of interest.

Instead of maximizing the sufficiency term of the IB, we formulate a new bottleneck method that minimizes deficiency. Deficiencies provide a principled way of approximating complex channels

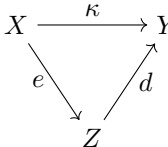

Figure 1: The bottleneck paradigm: The general idea of a bottleneck method is to first map an input $X$ to an intermediate representation $Z$, and then map $Z$ to an output $Y$. We call the mappings, resp., an encoder ($e$) and a decoder ($d$). In general, the true channel $\kappa$ is unknown, and only accessible through a set of training examples. We would like to obtain an approximation of $\kappa$.

by relatively simpler ones. The deficiency of a decoder with respect to the true channel between input and output variables quantifies how well any stochastic encoding at the decoder input can be used to approximate the true channel. Deficiencies have a rich heritage in the theory of comparison of statistical experiments (Blackwell, 1953; Le Cam, 1964; Torgersen, 1991). From this angle, the formalism of deficiencies has been used to obtain bounds on optimal risk gaps of statistical decision problems. As we show, the deficiency bottleneck minimizes a regularized risk gap. Moreover, the proposed method has an immediate variational formulation that can be easily implemented as a modification of the Variational Information Bottleneck (VIB) (Alemi et al., 2016). In fact, both methods coincide in the limit of single-shot Monte Carlo approximations. We call our method the Variational Deficiency Bottleneck (VDB).

Perfect maximization of the IB sufficiency corresponds to perfect minimization of the DB deficiency. However, when working over a parametrized model and adding the bottleneck regularizer, both methods have different preferences, with the DB being closer to the optimal risk gap. Experiments on basic data sets show that the VDB is able to obtain more compressed representations than the VIB while performing equally well or better in terms of test accuracy.

We describe the details of our method in Section 2. We elaborate on the theory of deficiencies in Section 3. Experimental results with the VDB are presented in Section 4.

## 2 THE VARIATIONAL DEFICIENCY BOTTLENECK (VDB)

Let $X$ denote an observation or *input* variable and $Y$ an *output* variable of interest. Let $p(x, y) = \pi(x)\kappa(y|x)$ be the true joint distribution, where the conditional distribution or *channel* $\kappa(y|x)$ describes how the output depends on the input. We consider the situation where the true channel is unknown, but we are given a set of $N$ independent and identically distributed (i.i.d.) samples $(x^{(i)}, y^{(i)})_{i=1}^N$ from $p$. Our goal is to use this data to learn a more structured version of the channel $\kappa$, by first "compressing" the input $X$ to an intermediate *representation* variable $Z$ and subsequently mapping the representation back to the output $Y$. The presence of an intermediate representation can be regarded as a bottleneck, a model selection problem, or as a regularization strategy.

We define a *representation model* and an *inference model* using two parameterized families of channels $e(z|x)$ and $d(y|z)$. We will refer to $e(z|x)$ and $d(y|z)$ as an *encoder* and a *decoder*. The encoder-decoder pair induces a model $\widehat{\kappa}(y|x) = \int d(y|z)e(z|x)dz$. Equivalently, we write $\widehat{\kappa} = d \circ e$.

Given a representation, we want the decoder to be as powerful as the original channel $\kappa$ in terms of ability to recover the output. The deficiency of a decoder $d$ w.r.t. $\kappa$ quantifies the extent to which any pre-processing of $d$ (by way of randomized encodings) can be used to approximate $\kappa$ (in the KL-distance sense). Let $\mathsf{M}(\mathcal{X}; \mathcal{Y})$ denote the space of all channels from $\mathcal{X}$ to $\mathcal{Y}$. We define the deficiency of $d$ w.r.t. $\kappa$ as follows.

**Definition 1.** Given the channel $\kappa \in \mathsf{M}(\mathcal{X}; \mathcal{Y})$ from $X$ to $Y$, and a decoder $d \in \mathsf{M}(\mathcal{Z}; \mathcal{Y})$ from some $Z$ to $Y$, the *deficiency of $d$ w.r.t. $\kappa$* is defined as

$$\delta^\pi(d, \kappa) = \min_{e \in \mathsf{M}(\mathcal{X}; \mathcal{Z})} D_{\mathrm{KL}}(\kappa \| d \circ e | \pi). \tag{1}$$

Here $D_{\mathrm{KL}}(\cdot \| \cdot | \cdot)$ is the conditional KL divergence (Csiszár & Körner, 2011), and $\pi$ is an input distribution over $\mathcal{X}$. The definition is similar in spirit to Lucien Le Cam's notion of weighted defi-

ciencies of one channel w.r.t. another (Le Cam, 1964; Torgersen, 1991, Section 6.2) and its recent generalization by Raginsky (2011).

We propose to train the model by minimizing the deficiency of $d$ w.r.t. $\kappa$ subject to a regularization that penalizes complex representations. The regularization is achieved by limiting the *rate* $I(Z; X)$, the mutual information between the representation and the raw inputs. We call our method the Deficiency Bottleneck (DB). The DB minimizes the following objective over all tuples $(e \in \mathsf{M}(\mathcal{X}; \mathcal{Z}), d \in \mathsf{M}(\mathcal{Z}; \mathcal{Y}))$:

$$\mathcal{L}_{DB}(e, d) := \delta^\pi(d, \kappa) + \beta I(Z; X). \tag{2}$$

The parameter $\beta \geq 0$ allows us to adjust the level of regularization.

For any distribution $r(z)$, the rate term admits a simple variational upper bound (Csiszár & Körner, 2011, Eq. (8.7)):

$$I(Z; X) \leq \int p(x, z) \log \frac{e(z|x)}{r(z)} \, dx \, dz. \tag{3}$$

Let $\hat{p}_{\text{data}}$ be the empirical distribution of the data (input-output pairs). By noting that $\delta^\pi(d, \kappa) \leq D_{\text{KL}}(\kappa \| d \circ e | \pi)$ for any $e \in \mathsf{M}(\mathcal{X}; \mathcal{Z})$, and ignoring (unknown) data-dependent constants, we obtain the following optimization objective which we call the Variational Deficiency Bottleneck (VDB) objective:

$$\mathcal{L}_{VDB}(e, d) := \mathbb{E}_{(x,y) \sim \hat{p}_{\text{data}}} \left[ -\log((d \circ e)(y|x)) + \beta D_{\text{KL}}(e(Z|x) \| r(Z)) \right]. \tag{4}$$

The computation is simplified by defining $r(z)$ to be a standard multivariate Gaussian distribution $\mathcal{N}(0, I)$, and using an encoder of the form $e(z|x) = \mathcal{N}(z|f_\phi(x))$, where $f_\phi$ is a neural network that outputs the parameters of a Gaussian distribution. Using the reparametrization trick (Kingma & Welling, 2013; Rezende et al., 2014), we then write $e(z|x)dz = p(\epsilon)d\epsilon$, where $z = f(x, \varepsilon)$ is a function of $x$ and the realization $\epsilon$ of a standard normal distribution. This allows us to do stochastic backpropagation through a single sample $z$. The KL term admits an analytic expression for a choice of Gaussian $r(z)$ and encoders. We train the model by minimizing the following empirical objective:

$$\frac{1}{N} \sum_{i=1}^{N} \left[ -\log\left(\frac{1}{M} \sum_{j=1}^{M} [d(y^{(i)}|f(x^{(i)}, \epsilon^{(j)}))]\right) + \beta D_{\text{KL}}(e(Z|x^{(i)}) \| r(Z)) \right]. \tag{5}$$

For training, we choose a mini-batch size of $N = 100$. For Monte Carlo estimates of the expectation inside the log, we choose $M = 3, 6, 12$ samples from the encoding distribution.

We note that the Variational Information Bottleneck (VIB) (Alemi et al., 2016) leads to a similar-looking objective function, with the only difference that the sum over $j$ is outside of the log. By Jensen's inequality, the VIB loss is an upper bound to our loss. If one uses a single sample from the encoding distribution (i.e., $M = 1$), the VDB and the VIB objective functions coincide.

The average log-loss and the rate term in the VDB objective equation 4 are the two fundamental quantities that govern the probability of error when the model is a classifier. For a discussion of these relations, see Appendix A.

## 3 BLACKWELL SUFFICIENCY AND CHANNEL DEFICIENCY

In this section, we discuss an intuitive geometric interpretation of the deficiency in the space of probability distributions over the output variable. We also give an operational interpretation of the deficiency as a deviation from Blackwell sufficiency (in the KL-distance sense). Finally, we discuss its relation to the log-loss.

### 3.1 DEFICIENCY AND DECISION GEOMETRY

We first formulate the learning task as a decision problem. We show that $\delta^\pi(d, \kappa)$ quantifies the gap in the optimal risks of decision problems when using the channel $d$ rather than $\kappa$.

Let $\mathcal{X}$, $\mathcal{Y}$ denote the space of possible inputs and outputs. In the following, we assume that $\mathcal{X}$ and $\mathcal{Y}$ are finite. Let $\mathbb{P}_{\mathcal{Y}}$ be the set of all distributions on $\mathcal{Y}$. For every $x \in \mathcal{X}$, define $\kappa_x \in \mathbb{P}_{\mathcal{Y}}$ as $\kappa_x(y) = \kappa(y|x)$, $\forall y \in \mathcal{Y}$. Nature draws $x \sim \pi$ and $y \sim \kappa_x$. The learner observes $x$ and *quotes* a distribution $q_x \in \mathbb{P}_{\mathcal{Y}}$ that expresses her uncertainty about the true value $y$. The quality of a quote $q_x$ in relation to $y$ is measured by an extended real-valued loss function called the *score* $\ell\colon \mathcal{Y} \times \mathbb{P}_{\mathcal{Y}} \to \overline{\mathbb{R}}$. For a background on such special kind of loss functions see, e.g., Grünwald et al., 2004; Gneiting & Raftery, 2007; Parry et al., 2012. Ideally, the quote $q_x$ should to be as close as possible to the true conditional distribution $\kappa_x$. This is achieved by minimizing the expected loss $L(\kappa_x, q_x) := \mathbb{E}_{y \sim \kappa_x} \ell(y, q_x)$, for all $x \in \mathcal{X}$. The score is called *proper* if $\kappa_x \in \arg\min_{q_x \in \mathbb{P}_{\mathcal{Y}}} L(\kappa_x, q_x)$. Define the *Bayes act* against $\kappa_x$ as the optimal quote

$$q_x^* := \underset{q_x \in \mathbb{P}_{\mathcal{Y}}}{\arg\min}\, L(\kappa_x, q_x).$$

If multiple Bayes acts exist then select one arbitrarily. Define the *Bayes risk* for the distribution $p_{XY}(x, y) = \pi(x)\kappa(y|x)$ as $R(p_{XY}, \ell) := \mathbb{E}_{x \sim \pi} L(\kappa_x, q_x^*)$. A score is *strictly proper* if the Bayes act is unique. An example of a strictly proper score is the log-loss function defined as $\ell_L(y, q) := -\log q(y)$. For the log-loss, the Bayes act is $q_x^* = \kappa_x$ and the Bayes risk is just the conditional entropy

$$R(p_{XY}, \ell_L) = \mathbb{E}_{x \sim \pi} \mathbb{E}_{y \sim \kappa_x} \big[ -\log q_x^*(y) \big] = \mathbb{E}_{x \sim \pi} \mathbb{E}_{y \sim \kappa_x} \big[ -\log \kappa_x(y) \big] = H(Y|X). \quad (6)$$

Given a representation $z \in \mathcal{Z}$ (output by some encoder), when using the decoder $d$, the learner is constrained to quote a distribution from a subset of $\mathbb{P}_{\mathcal{Y}}$. Let $C = \mathrm{conv}(\{d_z : z \in \mathcal{Z}\}) \subset \mathbb{P}_{\mathcal{Y}}$ be the convex hull of the points $\{d_z\}_{z \in \mathcal{Z}} \in \mathbb{P}_{\mathcal{Y}}$. The Bayes act against $d_z$ is

$$q_{x_Z}^* := \underset{q_x \in C}{\arg\min}\, \mathbb{E}_{y \sim \kappa_x} \big[ -\log q_x(y) \big]. \quad (7)$$

$q_{x_Z}^*$ has an interpretation as the *reverse I-projection* of $\kappa_x$ to the convex set of probability measures $C \subset \mathbb{P}_{\mathcal{Y}}$ (Csiszár & Matuš, 2003)[1]. We call the associated Bayes risk as the *projected Bayes risk* $R_Z(p_{XY}, \ell_L)$ and the associated conditional entropy as the *projected conditional entropy* $H_Z(Y|X)$,

$$R_Z(p_{XY}, \ell_L) = \mathbb{E}_{x \sim \pi} \mathbb{E}_{y \sim \kappa_x} \big[ -\log q_{x_Z}^*(y) \big] = H_Z(Y|X). \quad (8)$$

The gap in the optimal risks, $\Delta R := R_Z(p_{XY}, \ell_L) - R(p_{XY}, \ell_L)$ when making a decision based on an intermediate representation and a decision based on the input data is just the deficiency. This follows from noting that

$$\Delta R = H_Z(Y|X) - H(Y|X) = \sum_{x \in \mathcal{X}} \pi(x) \min_{q_x \in C \subset \mathbb{P}_{\mathcal{Y}}} D_{\mathrm{KL}}(\kappa_x \| q_x)$$

$$= \min_{e \in \mathsf{M}(\mathcal{X}; \mathcal{Z})} \sum_{x \in \mathcal{X}} \pi(x) D_{\mathrm{KL}}(\kappa_x \| d \circ e_x)$$

$$= \min_{e \in \mathsf{M}(\mathcal{X}; \mathcal{Z})} D_{\mathrm{KL}}(\kappa \| d \circ e | \pi) = \delta^\pi(d, \kappa). \quad (9)$$

$\Delta R$ vanishes if and only if the optimal quote against $d_z$, $q_{x_Z}^*$ matches $\kappa_x$ for all $x, y$. This gives an intuitive geometric interpretation of a vanishing deficiency in the space of distributions over $\mathcal{Y}$.

Given a decoder channel $d$, since $\delta^\pi(d, \kappa) \leq D_{\mathrm{KL}}(\kappa \| d \circ e | \pi)$ for any $e \in \mathsf{M}(\mathcal{X}; \mathcal{Z})$, the loss term in the VDB objective is a variational upper bound on the projected conditional entropy $H_Z(Y|X)$. However, this loss is still a lower bound to the standard cross-entropy loss in the VIB objective (Alemi et al., 2016), i.e.,

$$\mathbb{E}_{(x,y) \sim \hat{p}_{\mathrm{data}}} \left[ -\log d \circ e(y|x) \right] \leq \mathbb{E}_{(x,y) \sim \hat{p}_{\mathrm{data}}} \left[ \int -e(z|x) \log d(y|z) dz \right]. \quad (10)$$

This follows simply from the convexity of the negative logarithm function.

---

[1] Such a projection exists and is not necessarily unique. If nonunique, we arbitrarily select one of the minimizers as the Bayes act.

### 3.2 DEFICIENCY AS A KL-DISTANCE FROM INPUT-BLACKWELL SUFFICIENCY

In a seminal paper David Blackwell (1953) asked the following question: if a learner wishes to make an optimal decision about some target variable of interest and she can choose between two channels with a *common input* alphabet, which one should she prefer? She can rank the channels by comparing her optimal risks: she will *always* prefer one channel over another if her optimal risk when using the former is at most that when using the latter for any decision problem. She can also rank the variables purely probabilistically: she will *always* prefer the former if the latter is an *output-degraded* version of the former, in the sense that she can simulate a single use of the latter by randomizing at the output of the former. Blackwell showed that these two criteria are equivalent.

Very recently, Nasser (2017) asked the same question, only now the learner has to choose between two channels with a *common output* alphabet. Given two channels, $\kappa \in \mathsf{M}(\mathcal{X}; \mathcal{Y})$ and $d \in \mathsf{M}(\mathcal{Z}; \mathcal{Y})$, we say that $\kappa$ is *input-degraded* from $d$ and write $d \succeq_{\mathcal{Y}} \kappa$ if $\kappa = d \circ e$ for some $e \in \mathsf{M}(\mathcal{X}; \mathcal{Z})$. Stated in another way, $d$ can be reduced to $\kappa$ by applying a randomization at its input. Nasser (2017) gave a characterization of input-degradedness that is similar to Blackwell's theorem (Blackwell, 1953). We say, $d$ is *input-Blackwell sufficient* for $\kappa$ if $d \succeq_{\mathcal{Y}} \kappa$.

Input-Blackwell sufficiency induces a preorder on the set of all channels with the same output alphabet. In practice, most channels are uncomparable, i.e., one cannot be reduced to another by a randomization. When such is the case, the deficiency quantifies how far the true channel $\kappa$ is from being a randomization (by way of all input encodings) of the decoder $d$. See Appendix B for a brief summary of Blackwell-Le Cam theory.

### 3.3 DEFICIENCY AND THE LOG-LOSS

When $Y - X - Z$ is a Markov chain, the conditional mutual information $I(Y; X|Z)$ is the Bayes risk gap for the log-loss. This is apparent from noting that $I(Y; X|Z) = H(Y|Z) - H(Y|XZ) = H(Y|Z) - H(Y|X) = R(p_{ZY}, \ell_L) - R(p_{XY}, \ell_L)$. This risk gap is closely related to Blackwell's original notion of sufficiency. Since the log-loss is strictly proper, a vanishing $I(Y; X|Z)$ implies that the risk gap is zero for all loss functions. This suggests that minimizing the log-loss risk gap under a suitable regularization constraint is a potential recipe for constructing representations $Z$ that are *approximately sufficient* for $X$ w.r.t. $Y$, since in the limit when $I(Y; X|Z) = 0$ one would achieve $I(Y; Z) = I(Y; X)$. This is indeed the basis for the IB algorithm (Tishby et al., 1999) and its generalization, clustering with Bregman divergences (Banerjee et al., 2005; van Rooyen & Williamson, 2015; 2014).

One can also approximate a sufficient statistic by *minimizing deficiencies* instead. This is motivated from noting the following proposition.

**Proposition 2.** When $Y - X - Z$ is a Markov chain, $\delta^\pi(d, \kappa) = 0 \iff I(Y; X|Z) = 0$.

In general, for the bottleneck paradigms involving the conditional mutual information (IB) and the deficiency (DB), we have the following relationship:

$$\min_{e(z|x):\, I(Y;X|Z) \leq \epsilon} I(X; Z) \geq \min_{e(z|x):\, \delta^\pi(d,\kappa) \leq \epsilon} I(X; Z). \tag{11}$$

Our experiments corroborate that for achieving the same level of sufficiency, one needs to store less information about the input $X$ when minimizing the deficiencies than when minimizing the conditional mutual information.

## 4 EXPERIMENTS

We present some experiments on the MNIST dataset (LeCun & Cortes, 2010). Classification on MNIST is a very well studied problem. The main objective of our experiments is to evaluate the information-theoretic properties of the representations learned by the VDB and whether it can match the classification accuracy provided by other bottleneck methods.

For the encoder, we use a fully connected feedforward network with 784 input units–1024 ReLUs–1024 ReLUs–512 linear output units. The deterministic output of this network is interpreted as the vector of means and variances of a 256 dimensional Gaussian distribution. The decoder is a

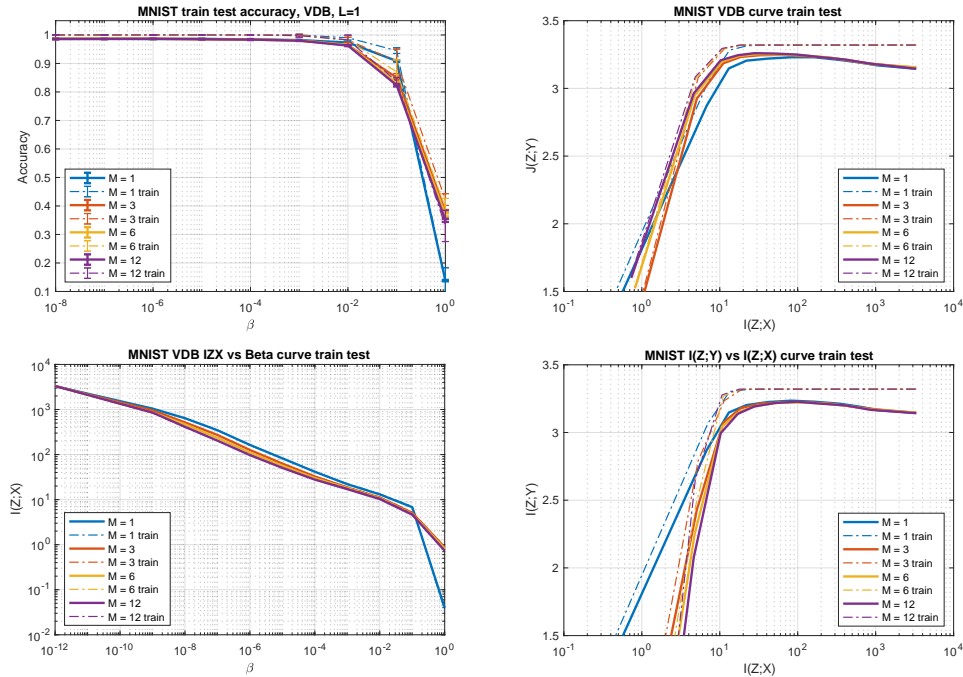

Figure 2: Effect of the regularization parameter $\beta$. The upper left panel shows the accuracy on train and test data after training the VDB for different values of $M$. Here, $M$ is the number of encoder output samples used in the training objective. $L$ is the number of encoder output samples used for evaluating the classifier. The upper right panel traces the *deficiency bottleneck curve* for different values of $\beta$ (see text). The curves are averages over 5 repetitions of the experiment. Each curve corresponds to one value of $M = 1, 3, 6, 12$. Notice the generalization gap for small values of $\beta$ (towards the right of the plot). The lower right panel plots the corresponding information bottleneck curve. The lower left panel plots the minimality term vs. $\beta$. Evidently, the levels of compression vary depending on $M$. Higher values of $M$ (our method) lead to a more compressed representation. For $M = 1$, the VDB and the VIB models coincide.

Table 1: Comparison of test accuracy values for different values of $\beta$ and $M$. $K$ is the size of the bottleneck and $L = 12$. We see a slight improvement in the test accuracies for higher values of $M$.

| $\beta$ | $K$ | $M$ | | | |
|---|---|---|---|---|---|
| | | 1 | 3 | 6 | 12 |
| $10^{-5}$ | 256 | 0.9869 | 0.9873 | 0.9885 | 0.9878 |
| | 2 | 0.9575 | 0.9678 | 0.9696 | 0.9687 |
| $10^{-3}$ | 256 | 0.9872 | 0.9879 | 0.9875 | 0.9882 |
| | 2 | 0.9632 | 0.9726 | 0.9790 | 0.9702 |

simple logistic regression model with a softmax layer. These are the same settings of the model used by Alemi et al. (2016). We implement the algorithm in Tensorflow and train for 200 epochs using the Adam optimizer.

As can be seen from the upper left panel in Figure 2, the test accuracy is stable with increasing $M$. Here, $M$ is the number of encoder output samples used in the training objective. We note that $M = 1$ is just the VIB model (Alemi et al., 2016). $L$ is the number of encoder output samples used for evaluating the classifier (i.e., we use $\frac{1}{L} \sum_{j=1}^{L} d(y|z^{(j)})$ where $z^{(j)} \sim e(z|x)$). Numerical values of the

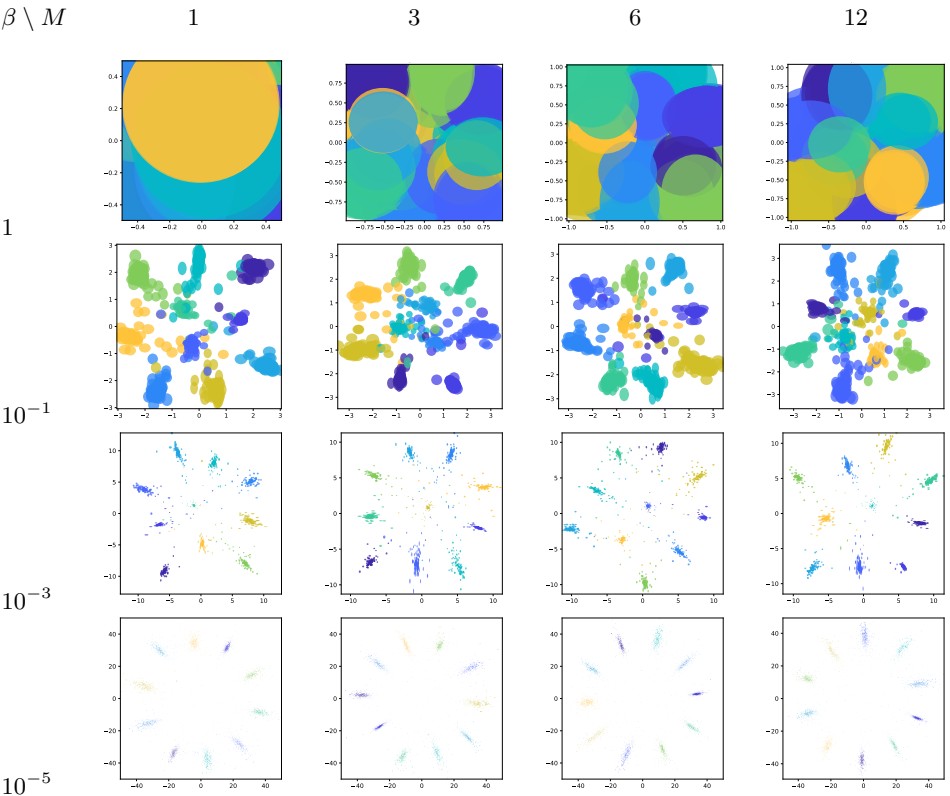

Figure 3: We trained the VDB on MNIST with the basic encoder given by a fully connected network with two hidden layers of ReLUs generating the means and variances of 2D independent Gaussian latent representation. Ellipses represent the posterior distributions of 1000 input images in latent space after training with $\beta = 10^0, 10^{-1}, 10^{-3}, 10^{-5}$ and $M = 1, 3, 6, 12$. Color corresponds to the class label.

test accuracies are provided in Table 1 for different values of $\beta$ and $M$. We see a slight improvement in the test accuracies for higher values of $M$. See Figure 5 for train and test accuracies for $L = 3$ and $L = 12$ in Appendix D. The traditional IB paradigm traces the mutual information $I(Z; Y)$ between representation and output (sufficiency) vs. the mutual information $I(Z; X)$ between representation and input (minimality), for different values of the regularization parameter $\beta$. This curve is called the *information bottleneck curve* (Tishby et al., 1999). In the case of the VDB, we define the corresponding sufficiency term as $J(Z; Y) := H(Y) - \mathbb{E}_{(x,y) \sim \hat{p}_{\text{data}}} \left[ -\log(\int d(y|z)e(z|x)\,dz) \right]$. Here $H(Y) = \log_2(10)$ is the entropy of the output which has 10 classes. In our method, "more informative" means "less deficient". The upper right panel in Figure 2 shows the *deficiency bottleneck curve* which traces $J(Z; Y)$ vs. $I(Z; X)$ for different values of $\beta$ at the end of training. For orientation, lower values of $\beta$ have higher values of $I(Z; X)$ (towards the right of the plot). For small values of $\beta$, when the effect of the regularization is negligible, the bottleneck allows more information from the input through the representation. In this case, $J(Z; Y)$ increases on the training set, but not necessarily on the test set. This is manifest in the gap between the train and test curves indicative of a degradation in generalization. For intermediate values of $\beta$, the gap is smaller for larger values of $M$ (our method). The lower right panel plots the corresponding information bottleneck curve. The lower left panel in Figure 2 plots the minimality term $I(Z; X)$ vs. $\beta$. We see that, for $\beta$ in the range between $10^{-8}$ and $10^{-4}$, for the same level of sufficiency, setting $M = 12$ consistently achieves more compression of the input compared to the setting $M = 1$. The dynamics of the information quantities during training are also interesting. We provide figures on these in Appendix D.

In order to visualize the representations, we also train the VDB on MNIST with a 2 dimensional representation. We use the same settings as before, with the only difference that the dimension of

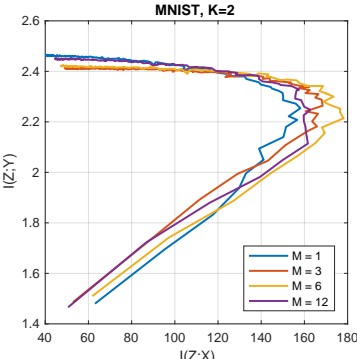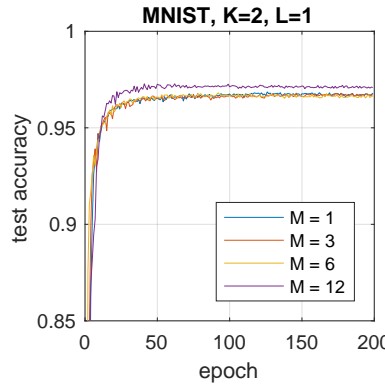

Figure 4: Learning curves for MNIST, where the encoder is a MLP of size $784$–$1024$–$1024$–$2K$, the last layer being a $K = 2$ dimensional diagonal Gaussian. The decoder is simply a softmax with $10$ classes. The left panel plots the mutual information between the representation and the class label, $I(Z; Y)$, against the mutual information between the representation at the last layer of the encoder and the input, $I(Z; X)$, as training progresses. The former increases monotonically, while the latter increases and then decreases. The right panel shows the test accuracy as training progresses.

the output layer of the encoder is $4$, with two coordinates representing the mean, and two a diagonal covariance matrix. The results are shown in Figure 3. For $\beta = 10^{-5}$, the representations are well separated, depending on the class. For related figures in the setting of unsupervised learning see Appendix E.

The learning dynamics of the mutual information and classification accuracy are shown in Figure 4. The left panel has an interpretation in terms of a phase where the model is mainly fitting the input-output relationship and hence increasing the mutual information $I(Z; Y)$, followed by a compression phase, where training is mainly reducing $I(Z; X)$, leading to a better generalization. The right panel shows the test accuracy as training progresses. Higher values of $M$ (our method) usually lead to better accuracy. An exception is when the number $L$ of posterior samples for classification is large.

## 5   DISCUSSION

We have formulated a bottleneck method based on channel deficiencies. The deficiency of a decoder with respect to the true channel between input and output quantifies how well a randomization at the decoder input (by way of stochastic encodings) can be used to simulate the true channel. The VDB has a natural variational formulation which recovers the VIB in the limit of a single sample of the encoder output. Experiments demonstrate that the VDB can learn more compressed representations while retaining the same discriminative capacity. The method has a statistical decision-theoretic appeal. Moreover, the resulting variational objective of the DB can be implemented as an easy modification of the VIB, with little to no computational overhead.

Given two channels that convey information about a target variable of interest, two different notions of deficiencies arise, depending on whether the target resides at the common input or the common output of the given channels. When the target is at the common output of the two channels, as is in a typical bottleneck setting (see Figure 1), our Definition 1 has a natural interpretation as a KL-divergence from input-Blackwell sufficiency (Nasser, 2017). Here sufficiency is achieved by applying a randomization at the input of the decoder with the goal of simulating the true channel. The notion of input-Blackwell sufficiency contrasts with Blackwell's original notion of sufficiency (Blackwell, 1953) in the sense that Blackwell's theory compares two channels with a common input. One can again define a notion of deficiency in this setting (see Appendix B for a discussion on deficiencies in the classical Blackwell setup). The associated channels (one from $\mathcal{Y}$ to $\mathcal{Z}$ and the other from $\mathcal{Y}$ to $\mathcal{X}$) do not however have a natural interpretation in a typical bottleneck setting. In contrast, the input-Blackwell setup appears to be much more intuitive in this context.

The more detailed view of information emerging from this analysis explains various effects and opens the door to multiple generalizations. In the spirit of the VDB, one can formulate a deficiency autoencoder as well (see sketch in Appendix E). On a related note, we mention that the deficiency is a lower bound to a quantity called the *Unique information* (Bertschinger et al., 2014; Banerjee et al., 2018a) (see details in Appendix C). An alternating minimization algorithm similar in spirit to the classical Blahut-Arimoto algorithm (Blahut, 1972) has been proposed to compute this quantity (Banerjee et al., 2018b). A deep neural network implementation of such an algorithm remains a challenge. In the limit $\beta \to 0$, the VDB is a step forward towards estimating the unique information. This might be of independent interest in improving the practicality of the theory of information decompositions.

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

APPENDIX

## A  MISCLASSIFICATION ERROR AND THE AVERAGE LOG-LOSS

In a classification task, the goal is to use the training dataset to learn a classifier $\widehat{\kappa}(y|x)$ that minimizes the *probability of error* under the true data distribution, defined as follows.

$$P_{\mathcal{E}}(\widehat{\kappa}) := 1 - \mathbb{E}_{(x,y)\sim p}\left[\widehat{\kappa}(y|x)\right]. \tag{12}$$

It is well known that the optimal classifier that gives the smallest probability of error is the Bayes classifier (Boucheron et al., 2005). Since we do not know the true data distribution we try to learn based on the empirical error. Directly minimizing the empirical probability of error over the training dataset is in general a NP-hard problem. In practice, one minimizes a surrogate loss function that is a convex upper bound on $P_{\mathcal{E}}$. A natural surrogate is the average log-loss function $\mathbb{E}_{(x,y)\sim p}\left[-\log\widehat{\kappa}(y|x)\right]$. When the model is $\widehat{\kappa} = d \circ e$, the following upper bounds are immediate from using Jensen's inequality.

$$\begin{aligned} P_{\mathcal{E}}(\widehat{\kappa}) &\leq 1 - \exp\left(-\mathbb{E}_{(x,y)\sim p}\left[-\log d \circ e(y|x)\right]\right) \\ &\leq 1 - \exp\left(-\mathbb{E}_{(x,y)\sim p}\mathbb{E}_{z\sim e(z|x)}\left[-\log d(y|z)\right]\right) \end{aligned} \tag{13}$$

The bound using the standard cross-entropy loss is evidently weaker than the average log-loss. A lower bound on the probability of error is controlled by a convex functional of the mutual information between the representation and the raw inputs $I(Z; X)$ (Vera et al., 2018, see, e.g., Lemma 4). The average log-loss and the rate term in the VDB objective equation 4 are two fundamental quantities that govern the probability of error.

## B  CLASSICAL THEORY OF COMPARISON OF CHANNELS

In this section, we discuss the classical theory of comparison of channels due to Blackwell (1953) and its extension by Le Cam (1964); Torgersen (1991) and more recently by Raginsky (2011).

Suppose that a learner wishes to predict the value of a random variable $Y$ that takes values in a set $\mathcal{Y}$. She has a set of actions $\mathcal{A}$. Each action incurs a loss $\ell(y, a)$ that depends on the true state $y$ of $Y$ and the chosen action $a$. Let $\pi_Y$ encode the learners' uncertainty about the true state $y$. The tuple $(\pi_Y, \mathcal{A}, \ell)$ is called a *decision problem*. Before choosing her action, the learner observes a random variable $X$ through a channel $\kappa \in \mathsf{M}(\mathcal{Y}; \mathcal{X})$. An ideal learner chooses a strategy $\rho \in \mathsf{M}(\mathcal{X}; \mathcal{A})$ that minimizes her expected loss or *risk* $R(\pi_Y, \kappa, \rho, \ell) := \mathbb{E}_{y\sim\pi_Y}\mathbb{E}_{a\sim\rho\circ\kappa_y}\ell(y, a)$. The *optimal risk* when using the channel $\kappa$ is $R(\pi_Y, \kappa, \ell) := \min_{\rho\in\mathsf{M}(\mathcal{X};\mathcal{A})} R(\pi_Y, \kappa, \rho, \ell)$.

Suppose now that the learner has to choose between $X$ and another random variable $Z$ that she observes through a second channel $\mu \in \mathsf{M}(\mathcal{Y}; \mathcal{Z})$ with common input $Y$. She can *always* discard $X$ in favor of $Z$ if, knowing $Z$, she can simulate a single use of $X$ by randomly sampling a $x' \in \mathcal{X}$ after each observation $z \in \mathcal{Z}$.

**Definition 3.** We say that $X$ is *output-degraded* from $Z$ w.r.t. $Y$, denoted $Z \sqsupseteq'_Y X$, if there exists a random variable $X'$ such that the pairs $(Y, X)$ and $(Y, X')$ are stochastically indistinguishable, and $Y - Z - X'$ is a Markov chain.

She can also discard $X$ if her optimal risk when using $Z$ is at most that when using $X$ for any decision problem. Write $Z \sqsupseteq_Y X$ if $R(\pi_Y, \kappa, \ell) \geq R(\pi_Y, \mu, \ell)$ for any decision problem. Blackwell (1953) showed the equivalence of these two relations.

**Theorem 4.** (Blackwell's Theorem) $Z \sqsupseteq'_Y X \iff Z \sqsupseteq_Y X$.

Write $\mu \sqsupseteq_{\mathcal{Y}} \kappa$ if $\kappa = \lambda \circ \mu$ for some $\lambda \in \mathsf{M}(\mathcal{Z}; \mathcal{X})$. If $\pi_Y$ has full support, then it easy to check that $\mu \sqsupseteq_{\mathcal{Y}} \kappa \iff Z \sqsupseteq'_Y X$ (Bertschinger & Rauh, 2014, Theorem 4).

The learner can also compare $\kappa$ and $\mu$ by comparing the mutual informations $I(Y; X)$ and $I(Y; Z)$ between the common input $Y$ and the channel outputs $X$ and $Z$.

**Definition 5.** $\mu$ is said to be *more capable* than $\kappa$, denoted $\mu \sqsupseteq^{mc}_{\mathcal{Y}} \kappa$, if $I(Y; Z) \geq I(Y; X)$ for all probability distribution on $\mathcal{Y}$.

It follows from the data processing inequality that $\mu \sqsupseteq_{\mathcal{Y}} \kappa \implies \mu \sqsupseteq_{\mathcal{Y}}^{mc} \kappa$. However, the converse implication is not true in general (Körner & Marton, 1975).

The converse to the Blackwell's theorem states that if the relation $Z \sqsupseteq'_Y X$ does not hold, then there exists a set of actions $\mathcal{A}$ and a loss function $\ell(y, a) \in \mathbb{R}^{\mathcal{Y} \times \mathcal{A}}$ such that $R(\pi_Y, \kappa, \ell) < R(\pi_Y, \mu, \ell)$. Le Cam introduced the concept of a *deficiency* of $\mu$ w.r.t. $\kappa$ to express this deficit in optimal risks (Le Cam, 1964) in terms of an approximation of $\kappa$ from $\mu$ via Markov kernels.

**Definition 6.** The *Le Cam deficiency of $\mu$ w.r.t. $\kappa$* is

$$\delta(\mu, \kappa) := \inf_{\lambda \in \mathsf{M}(\mathcal{Z}; \mathcal{X})} \sup_{y \in \mathcal{Y}} \|\lambda \circ \mu_y - \kappa_y\|_{\mathsf{TV}}, \tag{14}$$

where $\|\cdot\|_{\mathsf{TV}}$ denotes the total variation distance.

When the distribution of the common input to the channels is fixed, one can define a *weighted deficiency* (Torgersen, 1991, Section 6.2).

**Definition 7.** Given $Y \sim \pi_Y$, the *weighted Le Cam deficiency of $\mu$ w.r.t. $\kappa$* is

$$\delta^\pi(\mu, \kappa) := \inf_{\lambda \in \mathsf{M}(\mathcal{Z}; \mathcal{X})} \mathbb{E}_{y \sim \pi_Y} \|\lambda \circ \mu_y - \kappa_y\|_{\mathsf{TV}}. \tag{15}$$

Le Cam's *randomization criterion* (Le Cam, 1964) shows that deficiencies quantify the maximal gap in the optimal risks of decision problems when using the channel $\mu$ rather than $\kappa$.

**Theorem 8** (Le Cam (1964)). Fix $\mu \in \mathsf{M}(\mathcal{Y}; \mathcal{Z})$, $\kappa \in \mathsf{M}(\mathcal{Y}; \mathcal{Z})$ and a probability distribution $\pi_Y$ on $\mathcal{Y}$ and write $\|\ell\|_\infty = \max_{y,a} \ell(y, a)$. For every $\epsilon > 0$, $\delta^\pi(\mu, \kappa) < \epsilon$ if and only if $R(\pi_Y, \mu, \ell) - R(\pi_Y, \kappa, \ell) < \epsilon \|\ell\|_\infty$ for any set of actions $\mathcal{A}$ and any bounded loss function $\ell$.

Raginsky (2011) introduced a broad class of deficiency-like quantities using the notion of a generalized divergence between probability distributions that satisfies a monotonicity property w.r.t. data processing. The family of $f$-divergences due to Csiszár belongs to this class (Liese & Vajda, 2006).

**Definition 9.** The *$f$-deficiency of $\mu$ w.r.t. $\kappa$* is

$$\delta_f(\mu, \kappa) := \inf_{\lambda \in \mathsf{M}(\mathcal{Z}; \mathcal{X})} \sup_{y \in \mathcal{Y}} D_f(\kappa_y \| \lambda \circ \mu_y), \tag{16}$$

Many common divergences, such as the KL divergence, the reverse-KL divergence, and the total variation distance are $f$-divergences. When the channel $\mu$ is such that its output is constant, no matter what the input, the corresponding $f$-deficiency is called $f$-*informativity* (Csiszár, 1972). The $f$-informativity associated with the KL divergence is just the channel capacity which has a geometric interpretation as an "information radius" (Csiszár & Körner, 2011).

We can also define a weighted $f$-deficiency of $\mu$ w.r.t. $\kappa$.

**Definition 10.** The *weighted $f$-deficiency of $\mu$ w.r.t. $\kappa$* is

$$\delta_f(\mu, \kappa) := \inf_{\lambda \in \mathsf{M}(\mathcal{Z}; \mathcal{X})} D_f(\kappa_y \| \lambda \circ \mu_y | \pi_Y), \tag{17}$$

Specializing to the KL divergence, we have the following definition.

**Definition 11.** The *weighted output deficiency of $\mu$ w.r.t. $\kappa$* is

$$\delta_o^\pi(\mu, \kappa) := \min_{\lambda \in \mathsf{M}(\mathcal{Z}; \mathcal{X})} D_{\mathrm{KL}}(\kappa \| \lambda \circ \mu | \pi_Y), \tag{18}$$

where the subscript $o$ in $\delta_o^\pi$ emphasizes the fact that the randomization is at the *output* of the channel $\mu$.

Note that $\delta_o^\pi(\mu, \kappa) = 0$ if and only if $Z \sqsupseteq'_Y X$, which captures the intuition that if $\delta_o^\pi(\mu, \kappa)$ is small, then $X$ is *approximately output-degraded* from $Z$ w.r.t. $Y$. Using Pinsker's inequality, we have

$$\delta^\pi(\mu, \kappa) \leq \sqrt{\tfrac{\ln(2)}{2} \delta_o^\pi(\mu, \kappa)}. \tag{19}$$

## C  THE UNIQUE INFORMATION BOTTLENECK

In this section, we give a new perspective on the Information Bottleneck paradigm using nonnegative mutual information decompositions. The quantity we are interested in is the notion of *Unique information* proposed in (Bertschinger et al., 2014). Work in similar vein include (Harder et al., 2013) and more recently (Banerjee et al., 2018a) which gives an operationalization of the unique information.

Consider three jointly distributed random variables $Y$, $X$, and $Z$. $Y$ is the *target variable* of interest. The mutual information between $Y$ and $X$ can be decomposed into information that $X$ has about $Y$ that is *unknown* to $Z$ (we call this the *unique* information of $X$ w.r.t. $Z$) and information that $X$ has about $Y$ that is *known* to $Z$ (we call this the *shared* information).

$$I(Y;X) = \underbrace{\widetilde{UI}(Y;X\backslash Z)}_{\text{unique } X \text{ wrt } Z} + \underbrace{\widetilde{SI}(Y;X,Z)}_{\text{shared (redundant)}}. \tag{20}$$

Conditioning on $Z$ destroys the shared information but creates *complementary* or *synergistic* information from the interaction of $X$ and $Z$.

$$I(Y;X|Z) = \underbrace{\widetilde{UI}(Y;X\backslash Z)}_{\text{unique } X \text{ wrt } Z} + \underbrace{\widetilde{CI}(Y;X,Z)}_{\text{complementary (synergistic)}}. \tag{21}$$

Using the chain rule, the total information that the pair $(X,Z)$ conveys about the target $Y$ can be decomposed into four terms.

$$I(Y;XZ) = I(Y;X) + I(Y;Z|X) \tag{22}$$
$$= \widetilde{UI}(Y;X\backslash Z) + \widetilde{SI}(Y;X,Z) + \widetilde{UI}(Y;Z\backslash X) + \widetilde{CI}(Y;X,Z). \tag{23}$$

$\widetilde{UI}$, $\widetilde{SI}$, and $\widetilde{CI}$ are nonnegative functions that depend continuously on the joint distribution of $(Y,X,Z)$.

For completeness, we rewrite the information decomposition equations below.

$$I(Y;X) = \widetilde{UI}(Y;X\backslash Z) + \widetilde{SI}(Y;X,Z), \tag{24a}$$
$$I(Y;Z) = \widetilde{UI}(Y;Z\backslash X) + \widetilde{SI}(Y;X,Z), \tag{24b}$$
$$I(Y;X|Z) = \widetilde{UI}(Y;X\backslash Z) + \widetilde{CI}(Y;X,Z), \tag{24c}$$
$$I(Y;Z|X) = \widetilde{UI}(Y;Z\backslash X) + \widetilde{CI}(Y;X,Z), \tag{24d}$$

The unique information can be interpreted as either the conditional mutual information without the synergy, or as the mutual information without the redundancy.

When $Y - X - Z$ is a Markov chain, the information decomposition is

$$\widetilde{UI}(Y;Z\backslash X) = 0, \tag{25a}$$
$$\widetilde{UI}(Y;X\backslash Z) = I(Y;X|Z) = I(Y;X) - I(Y;Z), \tag{25b}$$
$$\widetilde{SI}(Y;X,Z) = I(Y;Z), \tag{25c}$$
$$\widetilde{CI}(Y;X,Z) = 0. \tag{25d}$$

The Information bottleneck (Tishby et al., 1999) minimizes the following objective

$$\mathcal{L}_{IB}(e) = I(Y;X|Z) + \beta I(X;Z), \tag{26}$$

over all encoders $e \in \mathsf{M}(\mathcal{X};\mathcal{Z}) : Y - X - Z$. Since $Y - X - Z$ is a Markov chain, the sufficiency term in the IB objective depends on the pairwise marginals $(Y,X)$ and $(Y,Z)$, while the minimality term depends on the $(X,Z)$-marginal. From equation 25b, it follows that one can equivalently write the IB objective function as

$$\mathcal{L}_{IB}(e) = \widetilde{UI}(Y;X\backslash Z) + \beta I(X;Z). \tag{27}$$

From an information decomposition perspective, the original IB is actually minimizing just the unique information subject to a regularization constraint. This is a simple consequence of the fact that the synergistic information $\widetilde{CI}(Y; X, Z) = 0$ (see equation 25d) when we have the Markov chain condition $Y - X - Z$. Hence, one might equivalently call the original IB as the *Unique information bottleneck*.

Appealing to classical Blackwell theory, Bertschinger et al. (2014) defined a nonnegative decomposition of the mutual information $I(Y; XZ)$ based on the idea that the unique and shared information should depend only on the pairwise marginals $(Y, X)$ and $(Y, Z)$.

**Definition 12.** Let $(Y, X, Z) \sim P, Y \sim \pi_Y$ and let $\kappa \in \mathsf{M}(\mathcal{Y}; \mathcal{X})$, $\mu \in \mathsf{M}(\mathcal{Y}; \mathcal{X})$ be two channels with the same input alphabet such that $P_{YX}(y, x) = \pi_Y(y)\kappa_y(x)$ and $P_{YZ}(y, z) = \pi_Y(y)\mu_y(z)$. Define

$$\Delta_P = \big\{ Q \in \mathbb{P}_{\mathcal{Y} \times \mathcal{X} \times \mathcal{Z}} \colon Q_{YX}(y, x) = \pi_Y(y)\kappa_y(x),$$

$$Q_{YZ}(y, z) = \pi_Y(y)\mu_y(z) \big\}, \tag{28a}$$

$$UI(Y; X \backslash Z) = \min_{Q \in \Delta_P} I_Q(Y; X | Z), \tag{28b}$$

$$UI(Y; Z \backslash X) = \min_{Q \in \Delta_P} I_Q(Y; Z | X), \tag{28c}$$

$$SI(Y; X, Z) = I(Y; X) - UI(Y; X \backslash Z), \tag{28d}$$

$$CI(Y; X, Z) = I(Y; X | Z) - UI(Y; X \backslash Z), \tag{28e}$$

where the subscript $Q$ in $I_Q$ denotes that joint distribution on which the quantities are computed.

The functions $UI$, $SI$, and $CI$ are nonnegative and satisfy equation 24. Furthermore, $UI$ and $SI$ depend on the marginal distributions of the pairs $(Y, X)$ and $(Y, Z)$. Only the function $CI$ depends on the full joint distribution $P$.

$UI$ satisfies the following intuitive property in relation to Blackwell's theorem 4.

**Proposition 13.** (Bertschinger et al., 2014, Lemma 6) $UI(Y; X \backslash Z) = 0 \iff Z \sqsupseteq'_Y X$.

Proposition 2 follows from noting that $\delta^\pi(d, \kappa) = 0 \iff UI(Y; X \backslash Z) = 0$ (Bertschinger et al., 2014, Theorem 22) and the fact that $UI(Y; X \backslash Z) = I(Y; X | Z)$ when $Y - X - Z$ is a Markov chain.

# D    ADDITIONAL FIGURES ON VDB EXPERIMENTS

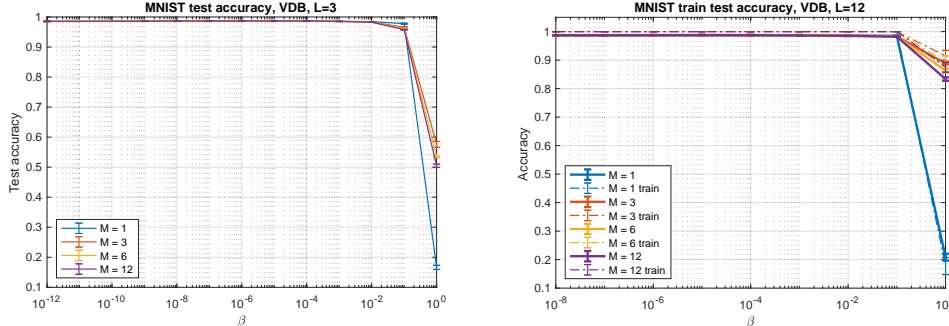

Figure 5: Train and test accuracy of the VDB for $L = 3$ and $L = 12$. Similar to Figure 2.

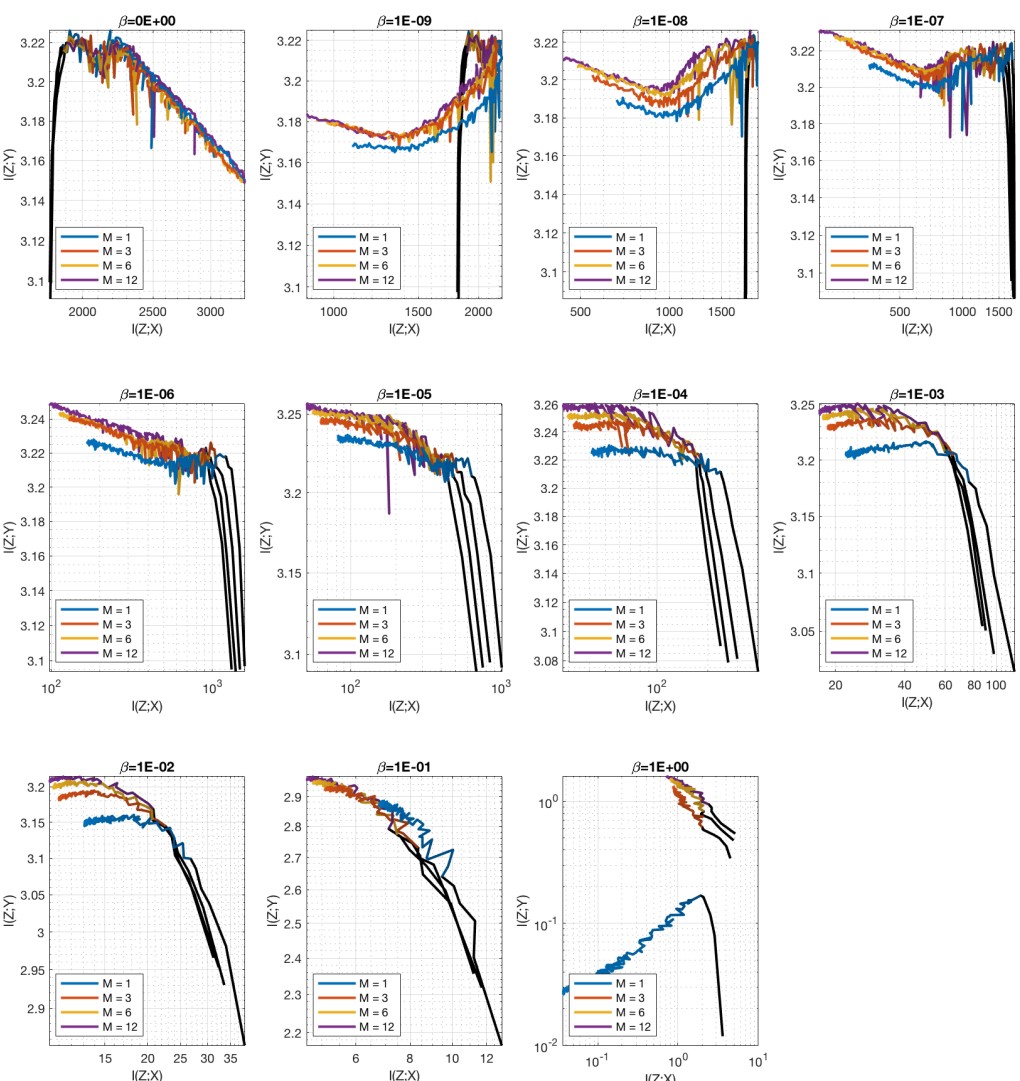

Figure 6: Evolution of the mutual information between representation and output vs. representation and input (values farther up and to the left are better) over 200 training epochs (dark to light color) on MNIST. The curves are averages over 20 repetitions of the experiment. At early epochs, training mainly effects fitting of the input-output relationship and an increase of $I(Z;Y)$. At later epochs, training mainly effects a decrease of $I(Z;X)$, which corresponds to the representation increasingly discarding information about the input. An exception is when the regularization parameter $\beta$ is very small. In this case the representation captures more information about the input, and longer training decreases $I(Z;Y)$, which is indicative of overfitting to the training data. Higher values of $M$ lead to the representation capturing more information about the target, while at the same time discarding more information about the input. $M = 1$ corresponds to the Variational Information Bottleneck.

# E   UNSUPERVISED REPRESENTATION LEARNING OBJECTIVES

Recent work on variational autoencoders (VAEs) has shown that optimizing the standard evidence lower-bound (ELBO) is not sufficient in itself for learning useful representations (Chen et al., 2016; Alemi et al., 2018; Zhao et al., 2017). Generalizing the ELBO by incorporating different bottleneck constraints (Zhao et al., 2017; Higgins et al., 2017; Rezende & Viola, 2018) has shown promise in learning better latent codes.

In this section, we discuss some preliminary results on an unsupervised version of the VDB objective. We discuss its relation to VAE objectives such as the $\beta$-VAE (Higgins et al., 2017) and the importance weighted autoencoder (IWAE) (Burda et al., 2015).

## E.1   $\beta$-VAE AND IWAE

A variational autoencoder (VAE) (Kingma & Welling, 2013; Rezende et al., 2014) is a directed probabilistic model that defines a joint density $p_\theta(x, z) = p_\theta(x|z)p(z)$ between some continuous latent variable $z$ and observed data $x$. $p(z)$ is chosen to be a simple prior distribution over the latents (e.g., isotropic unit Gaussian) and $p_\theta(x|z)$ is a decoder network that models the data generative process. Maximum likelihood estimation of the model parameters $\theta$ is in general intractable. VAEs instead maximize the evidence lower-bound (ELBO) by jointly training the decoder with an auxiliary encoder network $q_\phi(z|x)$, parameterized by $\phi$. The ELBO objective is

$$\mathcal{L}_{ELBO}(x) := \mathbb{E}_{z \sim q_\phi(z|x)}\left[\log \frac{p_\theta(x|z)p(z)}{q_\phi(z|x)}\right] \tag{29}$$

$$= \log p_\theta(x) - D_{\mathrm{KL}}(q_\phi(z|x)\|p_\theta(z|x)) \le \log p_\theta(x). \tag{30}$$

The ELBO is optimized by sampling from $q_\phi(z|x)$ using the reparameterization trick to obtain low variance Monte Carlo estimates of the gradient.

The ELBO is tight when $q_\phi(z|x)$ matches the true posterior $p_\theta(z|x)$. The tightness of the bound is coupled to the expressiveness of the encoder distribution. When $q_\phi(z|x)$ is chosen as a simple diagonal Gaussian, minimizing $D_{\mathrm{KL}}(q_\phi(z|x)\|p_\theta(z|x))$ encourages the model's posterior to be approximately factorial which limits the capacity of the model. The importance weighted autoencoder (IWAE) (Burda et al., 2015) addresses this issue. The key observation is that the unified ELBO objective in equation 29 is the log of a single (unnormalized) importance weight $\frac{p_\theta(x,z)}{q_\phi(z|x)}$ with a proposal density defined by $q_\phi(z|x)$. Using more samples from the proposal can only tighten the bound. The $M$-sample IWAE bound is

$$\mathcal{L}_M(x) := \mathbb{E}_{z^{1:M} \sim \prod_{i=1}^M q_\phi(z^{(i)}|x)}\left[\log\left(\frac{1}{M}\sum_{i=1}^M \frac{p_\theta(x, z^{(i)})}{q_\phi(z^{(i)}|x)}\right)\right]. \tag{31}$$

$\mathcal{L}_1$ is just the ELBO and $\lim_{M\to\infty}\mathcal{L}_M = \log p_\theta(x)$. The $M$-sample bound can alternatively be written as (Le et al., 2017)

$$\mathcal{L}_M(x) = \log p_\theta(x) - D_{\mathrm{KL}}(Q_{IS}\|P_{IS}) \le \log p_\theta(x), \tag{32}$$

where $Q_{IS}$ and $P_{IS}$ are, resp., proposal and target densities defined on an extended sample space.

$$Q_{IS}(z^{1:M}) = \frac{1}{M}\prod_{i=1}^M q_\phi(z^{(i)}|x), \; P_{IS}(z^{1:M}) = \frac{1}{M}\sum_{i=1}^M \frac{Q_{IS}(z^{1:M})}{q_\phi(z^{(i)}|x)}p_\theta(z^{(i)}|x). \tag{33}$$

Optimizing over this extended sample space allows for more flexible decoders and gives the IWAE additional degrees of freedom to model complex posteriors. The IWAE bound is in fact equivalent to the ELBO in expectation with a more complex proposal density $\tilde{q}_{IW}$ defined by importance reweighting (Bachman & Precup, 2015; Cremer et al., 2017; Naesseth et al., 2018).

$$\mathcal{L}_M(x) = \mathbb{E}_{z^{2:M}\sim\prod_{i=2}^M q_\phi(z^{(i)}|x)}\mathbb{E}_{z\sim\tilde{q}_{IW}(z|x,z^{2:M})}\left[\log\left(\frac{p_\theta(x, z)}{\tilde{q}_{IW}(z|x, z^{2:M})}\right)\right], \tag{34}$$

where the inner expectation is w.r.t. the unnormalized distribution $\tilde{q}_{IW}$ defined as follows.

$$\tilde{q}_{IW}(z|x, z^{2:M}) := M\tilde{w}q_\phi(z|x), \text{ where } \tilde{w} = \frac{\frac{p_\theta(x,z)}{q_\phi(z|x)}}{\sum_{j=1}^M \frac{p_\theta(x,z^{(j)})}{q_\phi(z^{(j)}|x)}}. \tag{35}$$

For $M = 1$, $\tilde{q}_{IW}(z|x) = q_\phi(z|x)$, and the unified ELBO objective admits the following decomposition.

$$\mathcal{L}_{ELBO}(x) = \mathcal{L}_1(x) = \mathbb{E}_{z \sim q_\phi(z|x)} \left[ \log p_\theta(x|z) \right] - D_{\mathrm{KL}}(q_\phi(z|x) \| p(z)). \tag{36}$$

The first term can be interpreted as an expected reconstruction cost and the second term as a regularizer (Kingma & Welling, 2013). For $M > 1$ however, the IWAE bound admits no such decomposition. As $M \to \infty$, $\tilde{q}_{IW}$ approaches the true posterior $p_\theta(z|x)$. However, the magnitude of the gradient w.r.t. the encoder parameters also decays to zero as more samples are used (Rainforth et al., 2018). This potentially limits the IWAE's ability to learn useful representations.

The $\beta$-VAE (Higgins et al., 2017; Burgess et al., 2018) augments the ELBO by incorporating a hyperparameter $\beta$ for the regularization term.

$$\mathcal{L}_\beta(x) = \mathbb{E}_{z \sim q_\phi(z|x)} \left[ \log p_\theta(x|z) \right] - \beta D_{\mathrm{KL}}(q_\phi(z|x) \| p(z)) \tag{37}$$

For $\beta = 1$, we recover the standard ELBO. Higgins et al. (2017) showed that when the prior $p(z)$ and the encoder $q_\phi(z|x)$ are chosen as diagonal Gaussians, reducing the capacity of the latent bottleneck by choosing $\beta > 1$ incentivizes the latent representations to be more disentangled. Higher values of $\beta$ also results in a more coherent latent space so that the reconstructions interpolate smoothly on latent traversals.

## E.2 Unsupervised Deficiency Bottleneck Objective

We now discuss some preliminary results on an unsupervised version of the VDB objective on the MNIST and Fashion-MNIST datasets. We consider a standard VAE model with the prior $p(z)$ and the encoder $q_\phi(z|x)$ parameterized by diagonal Gaussians. The encoder has two hidden layers with 200 units each. The decoder is a factorized Bernoulli parameterized by MLP's with two hidden layers with 200 units each. Using factorial decoder densities constrains the model to use the latent code to attain a high likelihood (Chen et al., 2016; Alemi et al., 2018). This is simple way to achieve a nonvanishing mutual information between the latent variable and the input. This is important in our setting since we are interested in learning a useful representation.

We train the model by minimizing the following unsupervised version of the VDB objective.

$$\frac{1}{N} \sum_{i=1}^{N} \left[ -\log\left(\frac{1}{M} \sum_{j=1}^{M} [p_\theta(x^{(i)}|f(x^{(i)}, \epsilon^{(j)}))]\right) + \beta D_{\mathrm{KL}}(q_\phi(Z|x^{(i)}) \| p(Z)) \right]. \tag{38}$$

We note that the $\beta$-VAE has a similar-looking training objective, with the only difference that the averaging w.r.t. to the posterior samples is outside the log. In particular, if $M = 1$, this is just the $\beta$-VAE objective. The objective in equation 38 also shares some superficial similarities with the IWAE objective for $\beta = 1$. Note however, as discussed in Section E.1, we cannot decompose the IWAE objective for $M > 1$. In particular, this implies we cannot trade-off reconstruction fidelity for learning more meaningful representations by incorporating bottleneck constraints. We have not explored if using more complex posteriors such as the $\tilde{q}_{IW}$ is possible in the bottleneck formulation.

For training, we choose a mini-batch size of $N = 100$ and draw $M = 1, 3, 6$ samples from the approximate posterior by using the reparameterization trick (Kingma & Welling, 2013). For our choice of Gaussian prior and encoder, the KL term can be computed and differentiated without estimation. We estimate the expected loss term using Monte Carlo sampling. Since the expectation is inside the log, higher values of $M$ increases the variance of gradient estimates. Also numerically handling the log requires some care. We used the log-sum-exp trick to compute the expectation. For values of $M$ beyond 12, we observe some degradation in the visualizations.

### E.3 VISUALIZATIONS

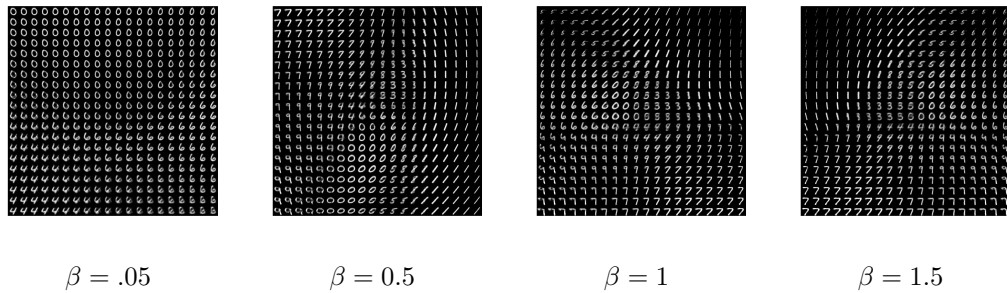

$\beta = .05$         $\beta = 0.5$         $\beta = 1$         $\beta = 1.5$

Figure 7: Sampling grids in latent space for $M = 6$ for different values of $\beta$ for the MNIST. Higher values of $\beta$ results in a more coherent latent space.

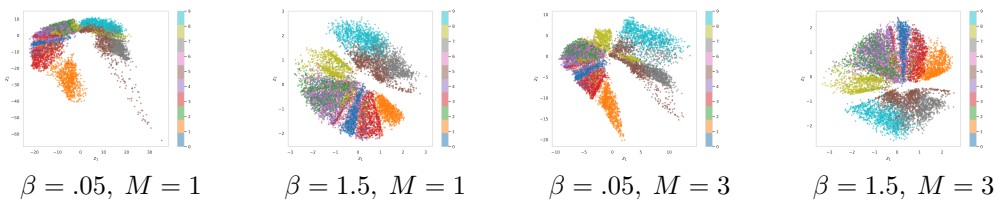

$\beta = .05, \ M = 1$      $\beta = 1.5, \ M = 1$      $\beta = .05, \ M = 3$      $\beta = 1.5, \ M = 3$

Figure 8: The latent space (mean values of the posterior for 5000 test examples) for the FMNIST for different values of $M$ and $\beta$. $M = 1$ corresponds to the $\beta$-VAE.

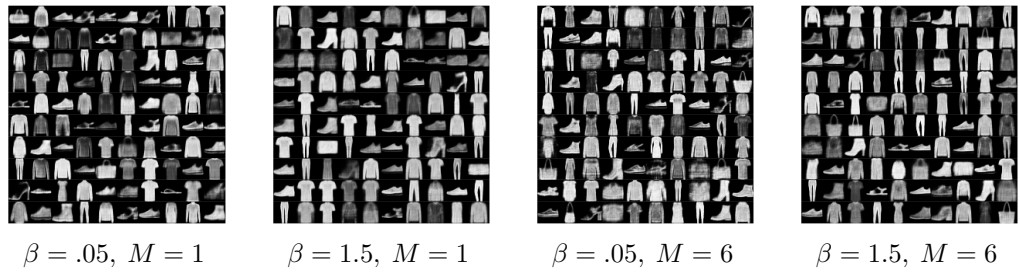

$\beta = .05, \ M = 1$      $\beta = 1.5, \ M = 1$      $\beta = .05, \ M = 6$      $\beta = 1.5, \ M = 6$

Figure 9: FMNIST reconstructions for different values of $M$ and $\beta$. At low values of $\beta$, we have good reconstructions. $M = 1$ corresponds to the $\beta$-VAE.

