# OpenReview forum: "The Variational Deficiency Bottleneck"
_ICLR.cc/2019/Conference_

### Official Review · AnonReviewer3 · 2018-11-02
**Good Writing, Comparisons Needed.**

**Rating:** 6
**Confidence:** 2

**Review:**

This paper introduces deficiency bottleneck for learning a data representation and represent  complicated channels using simpler ones. This problem has a natural variational form that can be easily implemented from VIB. Experiments show good performance comparing to VIB.

This paper is well-written and easy to read. The idea using KL divergence creating a deficiency channel to learn data representation is very natural. It is interesting that this formulation could be understood as minimizing a regularized risk gap of statistical decision problems, which justifies the usage of deficiency bottleneck (eq.9).

My biggest concern is the lack of comparison with other representation learning methods, which is a very well studied problem. However, it looks like authors only compared with VIB which is similar to the proposed method in terms of the objective function. For example, how does the method compare with (variants of) Variational Autoencoder? A discussion on this or some empirical evaluations would be nice.

---

> ### Author Response · Authors · 2018-11-27
> **Response to AnonReviewer3**
>
> Thank you for your comments!
>
> * We have enhanced Appendix E with a discussion on some variants of the VAE that generalize the standard evidence lower bound (ELBO) by incorporating different bottleneck constraints to learn better representations. In particular, we discuss how our unsupervised objective (p. 18, equation 38) relates to the beta-VAE and the importance weighted autoencoder (Appendix E.1, p. 17,18).
>
> * Please note that our unsupervised objective (p. 18, equation 38) contains the beta-VAE as a special case when we use only one sample from the encoding distribution (M=1). This means that we are naturally comparing with that method.
>
> * Appendix E.2 (p. 18) now discusses implementation details of the unsupervised objective. Finally, we have included some visualizations in Appendix E.3 for the MNIST and FMNIST datasets for different values of M and beta.
> We agree that more comparisons will be beneficial in investigating the properties of the proposed method. This is something we are actively working on.

---

### Official Review · AnonReviewer1 · 2018-11-06
**The paper presents a method of learning representations that is based on minimizing "deficiency" rather than optimizing for information sufficiency.**

**Rating:** 7
**Confidence:** 2

**Review:**

The paper presents a method of learning representations that is based on minimizing "deficiency" rather than optimizing for information sufficiency. While perfect optimization of the sufficiency term in IB is equivalent to minimizing deficiency, the thesis of the paper is that the variational upper bound on deficiency is easier to optimize, and when optimized produces
better (more compressed representations), while performing equally on test accuracy.



The paper is well written and easy to read. The idea behind the paper (optimizing for minimizing deficiency instead of sufficiency in IB) is interesting, especially because the variational formulation of DB is a generalization of VIB (in that VIB reduces to VDB for M=1). What takes away from the paper is that while perfect optimization of IB/sufficiency is equivalent to perfect optimization of DB, it is not clear what happens when perfection is not achieved. Further, the authors claim that DB is able to obtain more compressed representations (But is the goal a compressed representation, or an informative one?). The paper would also benefit from evaluation of the representation itself, and comparison to other non-information bottleneck based algorithms.

---

> ### Author Response · Authors · 2018-11-27
> **Response to AnonReviewer1**
>
> Thank you for your comments!
>
> * In our method, "``more informative" means "``less deficient".
> We have added a figure tracing the mutual information between representation and output I(Z;Y) vs. the minimality term I(Z;X) for different values of beta (see Figure 2, lower right panel), when training with our loss function. This is the usual information bottleneck curve. The deficiency bottleneck curve (Figure 2, upper right panel) traces the corresponding sufficiency term J(Z;Y) (which is just the entropy of the labels minus our loss) vs. I(Z;X) for different values of beta. The text now makes this more explicit (see p.7, first paragraph). Note that for M=1, J(Z;Y) = I(Z;Y). We can see that when training with our loss, we achieve approximately the same level of sufficiency (measured in terms of I(Z;Y)) while consistently achieving more compression (note the log ordinate for I(Z;X) in the lower left panel in Fig. 2) for a wide range of beta values.
>
> * We included two new figures plotting the representation for MNIST (p. 19, Figure 7) and Fashion-MNIST (p. 19, Figure 8) in Appendix E.3 for an unsupervised version of the VDB objective (p. 18, equation 38).

---

### Official Review · AnonReviewer2 · 2018-11-06
**New representation learning objective using Chanel Deficiency**

**Rating:** 5
**Confidence:** 5

**Review:**

This paper used the concept based on channel deficiency to derive a variational bound similar to variational information bottleneck. Theoretical analysis shows that this bound is an lower bound on the VIB objective. The empirical analysis shows it outperforms VIB in some sense.

I think this paper's contribution is rather theoretical than practical. The experiments section can be improved in the following aspect:
-  Figure 2 are hard to read for different M's. It would be better if the authors can show the exact accuracy numbers rather than the overlapped lines
- I(Z;Y) vs I(Z;X) graph is typically used in a VIB setting. In the paper's variational deficiency setting, although plotting I(Z;Y) vs I(Z;X) is necessary, it would be also helpful for the authors' to plot Deficiency vs I(Z;X), because this is what new objective is trading-off.
- Again, Figure 3, it is hard to see the benefits for increasing M from the visualizations for different clusterings.
- How do the paper estimate I(Z;Y) and I(Z;X) for plotting these figures? Does the paper use lower bound or some estimators? It should be made clear in the paper since these are non-trivial estimations.

Last comment is that, although the concept of `deficiency` in a bottleneck setting is novel, the similar idea for tighter bound of log likelihood has already been pursed in the following paper:

- Yuri Burda, Roger Grosse, Ruslan Salakhutdinov. Importance Weighted Autoencoders. ICLR 2016

It was kind of surprising that the authors did not cite this paper given the results are pretty much the same. It would also be helpful for the authors to do a comparison or connection section with this paper.

I like the paper in general, but given it still has some space for improvement, I would keep my decision as boarder line for now.

---

> ### Author Response · Authors · 2018-11-27
> **Response to AnonReviewer2**
>
> Thank you for your comments!
>
> * We included a table showing accuracy numbers for different values of beta and M (see p. 6, Table 1) for the latent bottleneck sizes K=256 (Figure 2) and K=2 (Figure 3).
>
> * In relation to the figures, we have improved these in the revision. We are added a figure tracing the mutual information between representation and output I(Z;Y) vs. the minimality term I(Z;X) for different values of beta (see Figure 2, lower right panel), when training with our loss function. This is the usual information bottleneck curve. This contrasts with the deficiency bottleneck curve (Figure 2, upper right panel) which traces the corresponding sufficiency term J(Z;Y) (which is just the entropy of the labels minus our loss) vs. I(Z;X) for different values of beta. Note that for M=1, J(Z;Y) = I(Z;Y). We apologize for the confusion. The text now makes this more explicit (see p.7, first paragraph).
>
> * In response to your question about how we estimate the mutual information. Yes, we minimize an upper bound on both the deficiency and the rate term (see p.3, equation 3 and discussion leading up to the VDB objective in equation 4). The estimation of this upper bound is simplified by our choice of the prior and the encoding distribution which are diagonal Gaussians. The KL term can be computed and differentiated without estimation. We estimate the expected loss term using Monte Carlo sampling. We draw samples from the encoder using the reparameterization trick and leverage automatic differentiation (in Tensorflow) to compute the gradients. Since the expectation is inside the log, gradient updates may have higher variance for larger values of M.
>
> Our model is a classifier and our loss term is a tighter bound on the misclassification error (bias) than the usual cross-entropy loss as in the VIB (see p. 12, equation 13). Trading bias for variance has been investigated in some recent works (see, e.g., Bamler, Robert, et al. "Perturbative black box variational inference." NIPS 2017). See last paragraph in p. 18 for the related discussion in the unsupervised setting.
>
> * In relation to the connection to IWAE, we have included a detailed discussion in Appendix E.1.
>
> The method is different from ours, except in the limiting case where M = 1 and beta =1, in which case it coincides with the beta-VAE and also with our method. After taking a close look, we make the following observations:
>
> For M > 1, the IWAE bound does not admit a decomposition like the standard ELBO (see equation 29 and 36) into a reconstruction loss term and a regularization term. In particular, this implies we cannot trade-off reconstruction fidelity for learning more meaningful representations by incorporating bottleneck constraints. See ensuing discussion in p.18 following equation 36.
> In contrast, our method has a tuning parameter beta.
>
> The IWAE bound is known to be equivalent to the ELBO in expectation with a more complex approximate posterior qIW (see p.17, equation 34 and 35 and references therein in Appendix E.1). For beta values other than 1, a naive trick would be to plant qIW in liue of qphi in equation 37 (p. 18) to get a beta-IWAE of sorts. It is not entirely clear however, why we would want to do so when modulating beta already suffices to tune the VAE towards autoencoding (low beta) or autodecoding behavior (high beta) depending on the requirement at hand. A similar argument goes in the direction of an "Importance weighted Variational Information Bottleneck". We have not explored if and how using more expressive posteriors such as the qIW (p. 17, equation 35) can help the supervised bottleneck formulations in VDB or VIB. This remains a scope for future study.
>
> We are now also citing the paper Yuri Burda, Roger Grosse, Ruslan Salakhutdinov. Importance Weighted Autoencoders. ICLR 2016.

---

### Public Comment · (anonymous) · 2018-11-26
**Related work**

The connection between the information bottleneck, compression, and deep neural networks is also described in   Shwartz-Ziv & Tishby 2017 [https://arxiv.org/abs/1703.00810].
That work should be referenced.

---

> ### Author Response · Authors · 2018-11-27
> **Reference added**
>
> Thank you for the comment!
>
> We have now included the said reference.

---

### Meta-Review · Area_Chair1 · 2018-12-14
**Valid theory, but quite close to existing work**

**Confidence:** 3
**Recommendation:** Reject

**Metareview:**

Strengths:  The paper presents an alternative regularized training objective for supervised learning that has a reasonable theoretical justification.  It also has a simple computational formula.

Weaknesses:
The experiments are minimal proofs of concept on MNIST and fashion MNIST, and the authors didn't find an example where this formulation makes a large difference.  The resulting formula is very close to existing methods.  Finally the paper is a bit dense and the intuitions we should gain from this theory aren't made clear.

Points of contention:
One reviewer pointed out the close connection of the new objective to IWAE, and the authors added a discussion of the relation and showed that they're not mathematically equivalent.  However, as far as I can tell they're almost identical in purpose:  As k -> \infty in IWAE, the encoder ceases to matter.  And as M -> \infty in VDB, we take the max over all encoders.  Could the method proposed in this paper lead to an alternative to IWAE in the VAE setting?

Consensus:
Consensus wasn't reached, but the "7" reviewer did not appear to have put much though into their review.